# Genetic Mapping for Leaf Shape and Leaf Size in Non-Heading Chinese Cabbage by a RIL Population

**Tianzi Zhao [1], Aimei Bai [1], Xinya Wang [1], Feixue Zhang [1,2], Miaomiao Yang [1], Yuhui Wang [1], Tongkun Liu [1], Xilin Hou [1] and Ying Li [1,*]**

[1] State Key Laboratory of Crop Genetics & Germplasm Enhancement and Utilization, Engineering Research Center of Germplasm Enhancement and Utilization of Horticulture Crops, Ministry of Education, College of Horticulture, Nanjing Agricultural University, Nanjing 210095, China; 2021104057@stu.njau.edu.cn (T.Z.)

[2] Institute of Horticulture, Huzhou Academy of Agricultural Sciences, Huzhou 313000, China

\* Correspondence: yingli@njau.edu.cn; Tel: +86-025-84395756

**Abstract:** Leaves are the predominant photosynthetic and edible organs in non-heading Chinese cabbage (*Brassica campestris* ssp. *chinensis*, NHCC), contributing significantly to yield, appearance, and desirability to consumers. However, the genetic basis of leaf shape and size in non-heading Chinese cabbage remains unclear. In this study, we developed a RIL population using 'Maertou', with slender leaves and narrow petioles, and 'Suzhouqing', with oval leaves and wide petioles, to construct a genetic linkage map and detect QTLs. To obtain stable and reliable QTLs, the 11 leaf-related traits, including the leaf length, leaf width, and fresh weight of the lamina and petiole and the thickness of petiole was observed on two locations—while the leaf shape, petiole shape, index of lamina/petiole length, and index of petiole fresh weight were calculated based on 7 leaf-related traits. QTL mapping illustrated that a total of 27 QTLs for leaf-related traits were preliminarily detected. The candidate genes were annotated and several genes involved in leaf development and leaf shape appeared in the overlapping regions of multiple loci, such as *KRP2*, *GRF4*, *ARGOS*, and *SAUR9*. This study lays the foundation for further exploration of the genetic mechanisms and development of effective molecular markers for leaf shape and size in NHCC.

**Keywords:** non-heading Chinese cabbage; genetic map; quantitative trait loci; leaf shape; leaf size





## 1. Introduction

Non-heading Chinese cabbage (*Brassica campestris* ssp. *chinensis*, NHCC), a biennial herb, is an important species of the genus *Brassica* in *Brassicaceae* and an important leafy vegetable widely cultivated in China. Compared with Chinese cabbage, NHCC has flat leaves and does not form a leaf head, and can be divided into six varieties that display morphological and genetic diversity in terms of laminas and petioles [1]. Leaf shape, petiole shape, and leaf weight in NHCC are the most important leaf features to consumers. Meanwhile, these features also contribute to the architectural construction and biomass production through photosynthesis [2]. Therefore, revealing genetic mechanisms for leaf shape and size is crucial for breeding novel germplasms.

A genetic linkage map is a powerful tool for the identification of QTL. On the basis of various molecular markers, such as SSR [3], SRAP [4], EST-based SNP [5], and polymorphic markers on corresponding candidate genes [6], genetic linkage maps were constructed and used for detecting the amounts of QTL for leaf lobes, leaf color, leaf shape, leaf size, and flowering time. Apart from genotyping by traditional markers, the wide application of high-throughput sequencing allows the rapid selection of single nucleotide polymorphism (SNP) and insertion/deletion (InDel) markers. QTL for leaf color [7] and 11 leaf-related traits [8] were detected by a linkage map based on SNP makers obtained by genotyping-by sequencing (GBS). Bulked segregant analysis-sequencing (BSA-seq) and GradedPool-sequencing (GPS) were also used for QTL mapping and could be more efficient for the

quality traits controlled by single or two genes. A candidate gene for rolled leaves and short petioles was detected using BSA-seq and RNA-seq in soybeans [9]. The two methods are efficient for quality traits but cannot detect the minor QTL. It is more valuable to construct a genetic linkage map for populations with multiple variations, which can be reusable for different traits in the same or corresponding populations and minor QTL. For example, a linkage map of a $F_2$ population was constructed to identify QTL for bolting traits in *Brassica rapa* [10] and then a major QTL for leaf lobes was detected based on the same linkage map using a corresponding $F_{2:3}$ population [11].

Compared with segregating populations, immortal populations such as doubled haploid (DH) and recombinant inbred line (RIL) mapping populations are appropriate for analyzing complex agronomic traits [3]. Because of the genetic uniformity and stability of each line, the permanent populations can be replicated in multiple environments, alleviating the environmental effects, enabling researchers to harvest more accurate phenotypic data and detecting QTL associated with growth conditions and treatments, making the result more reliable. For example, using a population comprising inbred lines, the physical data of plant height and branch number at low-phosphorus and sufficient phosphorus supply growth conditions were collected and significant SNPs and candidate genes for plant height at a low-phosphorus supply were detected [12].

Leaf development is affected by multiple factors and its molecular regulatory network shows complexity. Cell proliferation and cell expansion are the main driving factors for leaf development. The CYCLINS (CYCs) complexed with CYCLIN-DEPENDENT KINASES (CDKs), the E2F/DIMERISATION PROTEIN (DP) transcriptional regulatory proteins, KIP-RELATED PROTEIN/INTERACTOR OF CDKs (KRP/ICK), and SIAMESE/SIAMESE-RELATED (SIM/SMR) proteins are the main core cell cycle proteins which ensure correct transmission of the genetic information and cell cycle, controlling the leaf size. Six important gene regulatory modules affect leaf growth and leaf size through regulating cell proliferation: ubiquitin receptor DA1–ENHANCER OF DA1 (EOD1) [13], GROWTH REGULATING FACTOR (GRF)–GRF-INTERACTING FACTOR (GIF) [14], SWITCH/SUCROSE NON-FERMENTING (SWI/SNF) [15], gibberellin (GA)–DELLA [16], KLU [17], and PEAPOD (PPD) [18]. Plant hormones especially auxin and gibberellins, with genes involved in biosynthesis and signaling pathways, are necessary for plant development and play a substantial role in the regulation of leaf size. It has been demonstrated that the number of leaf veins seems to have a strong positive correlation with the leaf size and leaf shape [16–18]. In addition, epigenetic and micro RNA also involve in regulation of leaf size [19,20].

Many studies of genetic mapping on leaf-related traits have been conducted in *Brassica rapa* and an amount of QTL for leaf size, leaf shape, and leaf weight were detected [3,6,8,21,22]. Some pleiotropic QTL were colocalized with several traits such as leaf length, leaf width, leaf shape, and petiole shape, suggesting strong correlations among leaf-related traits. Candidate genes for leaf-related traits were identified, such as *ASYMMETRIC LEAVES 1* (*BrAS1*), *LONGIFOLIA 1* (*BrLNG1*), *HASTY 1* (*BrHST1*), *PIN-FORMED 1* (*BrPIN1*), and *BrKRP2*, and colocalized with total leaf length and leaf width, while *BrAS1* and *BrLNG1* were associated with leaf shape. *BrGRF5* was detected as the candidate genes for length of lamina and petiole in *Brassica rapa* [6].

In this study, we constructed a genetic linkage map using an RIL population with 144 lines, which were developed by two varieties of non-heading Chinese cabbage. To obtain stable and reliable QTL, 11 leaf-related traits was collected in two locations. A total of 27 significant QTLs in 7 chromosomes were detected and further candidate genes regulating leaf morphology and development were annotated. Our findings facilitated the dissection of further exploration of the genetic mechanisms for leaf development and molecular breeding in NHCC.

## 2. Materials and Methods

### 2.1. Plant Materials

Two advanced inbred lines, 'Maertou' and 'Suzhouqing' were used as the female parent and the male parent to develop a F$_{7:8}$ recombinant inbred lines (RILs) population consisting of 144 lines. 'Maertou' (*Brassica campestris* ssp. *chinensis* Makino *var. multiceps* Hort), a landrace in Nantong, Jiangsu province, has slender blades and short and narrow petioles. 'Suzhouqing' (*Brassica campestris* ssp. *chinensis* Makino *var. communis* Tsen et Lee) was a landrace in Suzhou, Jiangsu province and showed oval leaves and wide, thick petioles.

Parental lines and RIL populations were planted in the Baima Research Station of Nanjing Agricultural University (31°35′ N and 119°09′ E) in Jiangsu province and at the Huzhou Experimental Station in Huzhou, Zhejiang Province (120°05′ E, 30°54′ N). Plants were sown in September 2021 and phenotype data were evaluated at about 98 days after germination. Experiments were designed in a randomized complete block design (RCBD).

### 2.2. Phenotypic Data Collection and Analysis

Three plants as biological replicates for each line were measured for their leaf-related traits. The fourth leaf of each plant was selected for measuring. For each leaf, the length of the lamina (LL) and petiole (PL), the width of lamina (LW) and petiole (PW), the fresh weight of the lamina (LFW) and petiole (PFW), and the thickness of the petiole (PT) were observed (Table 1). As Figure 1 showed, the length and fresh weight of the petioles were measured from the base of petiole to the bottom of the lamina; whereas the length and fresh weight of petiole were measured from the bottom of the lamina to the tip of the lamina. The width of the leaf was measured at the widest point. We described the leaf shape (LS) using the ratio of total leaf length to lamina width. The petiole shape (PS) was the ratio of petiole length to petiole width. The ratio of lamina length to petiole length (LPLI) was used to describe the length of lamina relative to petiole. The index of petiole fresh weight (PFWI) was evaluated by the ratio of petiole fresh weight to total leaf fresh weight.

Estimation of the best linear unbiased predictors (BLUPs) of each trait were performed with the *R/lme4*(1.1-34 version) package [23], with all factors (genotypes, locations, and the interactions of genotypes and locations) as random factors and their variances were extracted for a heritability calculation. Broad-sense heritability ($H^2$) was estimated as follows: $H^2 = \sigma^2_g/(\sigma^2_g + \sigma^2_{GL}/L + \sigma^2_e/L \times R)$, where $\sigma^2_g$, $\sigma^2_{GL}$, and $\sigma^2_e$ were the variance components estimated from the ANOVA for the genotypic, genotype × location variances, error, respectively, with *R* as the number of replicates and *L* as the number of locations. Spearman's rank correlation coefficient of BLUPs values of leaf-related traits was calculated using Chiplot (https://www.chiplot.online/) accessed on 15 March 2024.

**Table 1.** Description of leaf-related traits in *Brassica campestris*.

|  | Trait Name | Trait Description | Units |
|---|---|---|---|
| LL | Lamina length | Length from the bottom of the lamina to the tip of the lamina | cm |
| LW | Lamina width | Width of lamina at the widest point | cm |
| PL | Petiole length | Length from the base of the petiole to the bottom of the lamina | cm |
| PW | Petiole width | Width of petiole at the widest point | cm |
| LFW | Lamina fresh weight | Fresh weight from the bottom of the lamina to the tip of the lamina | g |
| PFW | Petiole fresh weight | Fresh weight from the base of the petiole to the bottom of the lamina | g |
| PT | Petiole thickness | Thickness of petiole | cm |
| LS | Leaf shape | Ratio of total leaf length to lamina width | |
| LPLI | Index of lamina/petiole length | Ratio of lamina length to petiole length | |
| PS | Petiole shape | Ratio of petiole length to petiole width | |
| PFWI | Index of petiole fresh weight | Ratio of petiole fresh weight to total leaf fresh weight | |

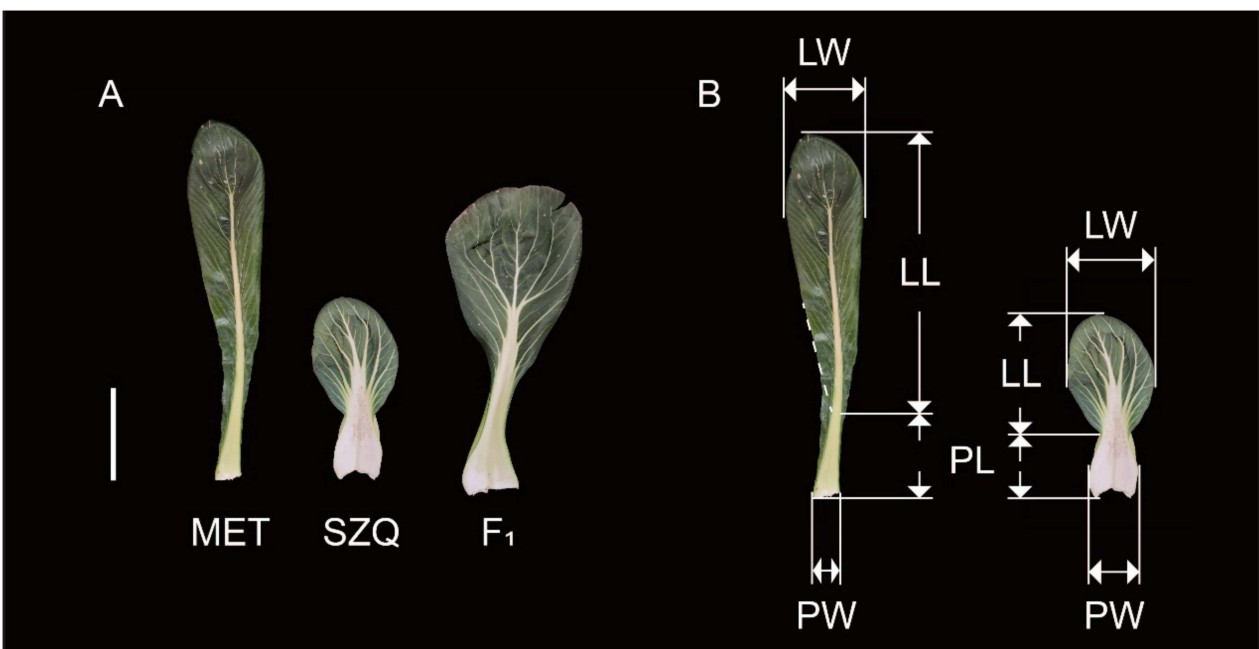

**Figure 1.** Morphological characteristics of leaves in the two parental lines ('Maertou' and 'Suzhouqing') and description of traits measured. (**A**): Leaf morphology of 'Maertou' (left), 'Suzhouqing' (middle) and F$_1$ hybrid (right) at the vegetative stage; bar = 10 cm. (**B**): Description of traits measured in this study. All traits and their descriptions are listed in Table 1. See Table 1 for trait abbreviation. MET—parental lines 'Maertou'; SZQ—parental lines 'Suzhouqing'; F$_1$—the F$_1$ hybrid.

*2.3. Construction of Genetic Map and QTL Analysis by Resequencing*

The genomic DNA of parental lines and the RIL population were extracted from their leaves using a genomic DNA extraction kit (TIANGEN, Beijing, China). The purity and concentration of the sample were determined by NanoPhotometer® (IMPLEN, Westlake Village, CA, USA) and a Qubit® 3.0 Flurometer (Life Technologies, Carlsbad, CA, USA). Fragmentase and other reagents were added to the DNA sample for DNA fragmentation. Then, the fragment size was repaired at the end and a tail was added and connected to the sequencing connector. Finally, the DNA library was obtained by PCR enrichment. After measuring the DNA concentration and the insert size of library, the library effective concentration was quantified accurately. The quantified libraries were sequenced using the Illumina Novaseq 6000 S4 platform (Illumina, San Diego, CA, USA) and 150 bp paired-end reads were generated. The construction of library and sequencing were performed at Annoroad Gene Tech (Beijing, China, https://www.annoroad.com/, accessed on 1 March 2022).

The clean reads of each line were aligned against the *Brassica campestris* genome (NHCC001; Version 1.0) [24] and downloaded from the non-heading Chinese cabbage and watercress database (http://tbir.njau.edu.cn/NhCCDbHubs/, accessed on 15 March 2024) [25] using the Burrows–Wheeler Aligner (BWA, v0.7.17-r1198-dirty) [26]. The SAMtools pipelines were used for further analysis [27]. SNP and InDel variations were selected using a Genome Analysis Toolkit (GATK) pipeline [28]. Low-quality SNPs/InDels were filtered with the criteria (a base quality value <40 and deep <8×). The SNP makers followed an *aa* × *bb* pattern were selected for the construction of a linkage map. The markers with missing rate exceeded 10% in RIL populations and the markers with distorted segregations were discarded.

The *mstmap* function of the *R/ASMap* (1.0-6 version) package [29] was used for estimating genetic distance. A linkage map was constructed using the *Kosambi* function in *R/qtl* (1.60 version) and *R/qtl2* packages (0.32 version) [30,31]. QTL analysis was performed by the *R/qtl* based on the composite interval mapping (CIM) method and multiple QTL

mapping (MQM). A total of 1000 permutations were adopted for calculating the LOD threshold at the 0.05 level. The additive effect and phenotypic variance of peak marker for each QTL were estimated by the function *fitqtl*. The intervals for QTL were supported by a 1.5 LOD drop interval.

### 2.4. Candidate Gene Analysis

All putative genes with their annotation information in the target interval were obtained from the *Brassica campestris* 'NHCC001' genome databases (http://tbir.njau.edu.cn/NhCCDbHubs/, accessed on 20 March 2024) [24]. The gene description was obtained from TAIR database (https://www.arabidopsis.org/, accessed on 20 March 2024) using BLASTp with a cutoff E value at '1 × 10⁻⁴'.

## 3. Results

### 3.1. Phenotypic Variations and Genetic Analysis of Leaf-Related Traits in NHCC

The two parental lines showed significant differences in leaf traits (Figures 1A and 2 and Table S1): 'Maertou' showed more slender leaves and more narrow but longer petioles and 'Suzhouqing' exhibited oval leaves and wider, thicker petioles. The LL, PL, LFW, LS, and PS of 'Suzhouqing' were 65.96%, 65.34%, 60.48%, 55.95%, and 37.03% of in 'Maertou', respectively; the LW, PW, PFW, PT, and PFWI of 'Maertou' were 86.85%, 57.19%, 52.63%, 42.66%, and 63.43% of in 'Suzhouqing', respectively. The F₁ hybrid exhibited oval leaves, wide and longer petioles, similar with 'Suzhouqing'. F₁ showed transgressive inheritance in PL and LPLI and exhibited intermediate traits between parents.

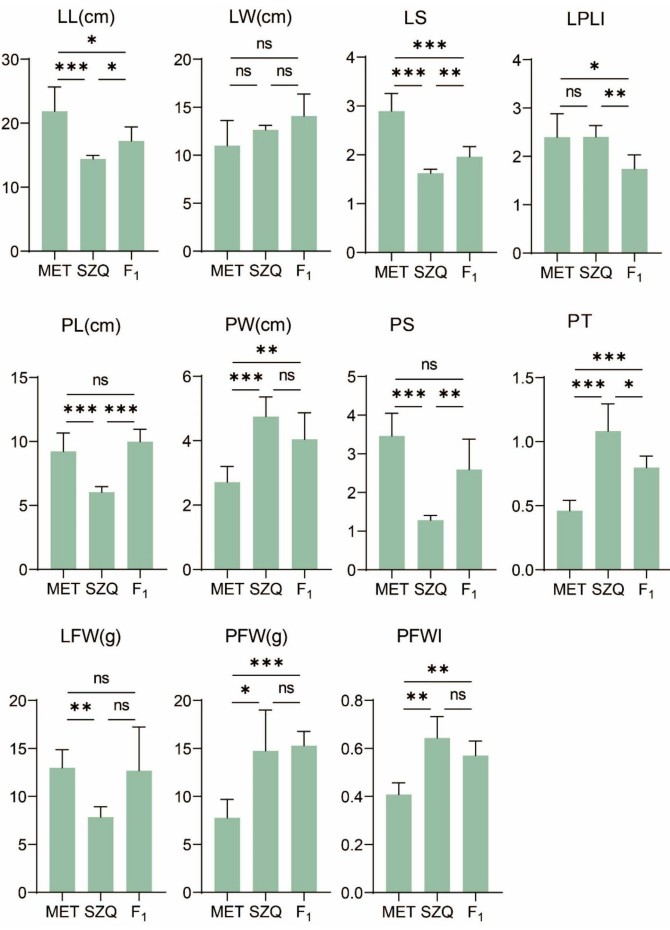

**Figure 2.** Comparison of 11 leaf-related traits between 'Maertou', 'Suzhouqing', and F₁. See Table 1 for trait abbreviation. MET—parental lines 'Maertou'; SZQ—parental lines 'Suzhouqing'; F₁—the F₁ hybrid. * $p \leq 0.05$, ** $p \leq 0.01$, *** $p \leq 0.001$, ns: not significant difference.

All leaf-related traits in the RIL population exhibited normal distribution (Figure 3 and Figure S1), with LW, PW, LPLI, and LFW showing transgressive segregation, suggesting that leaf-related traits were controlled by polygenes. The broad-sense heritability ($H^2$) for eleven leaf-related traits was 0.19–0.56 (Table 2), and the heritability indicated the traits were controlled by genetics and environment. To obtain comprehensive loci, the phenotypic data and BLUP value of 11 leaf-related traits were used for further analysis.

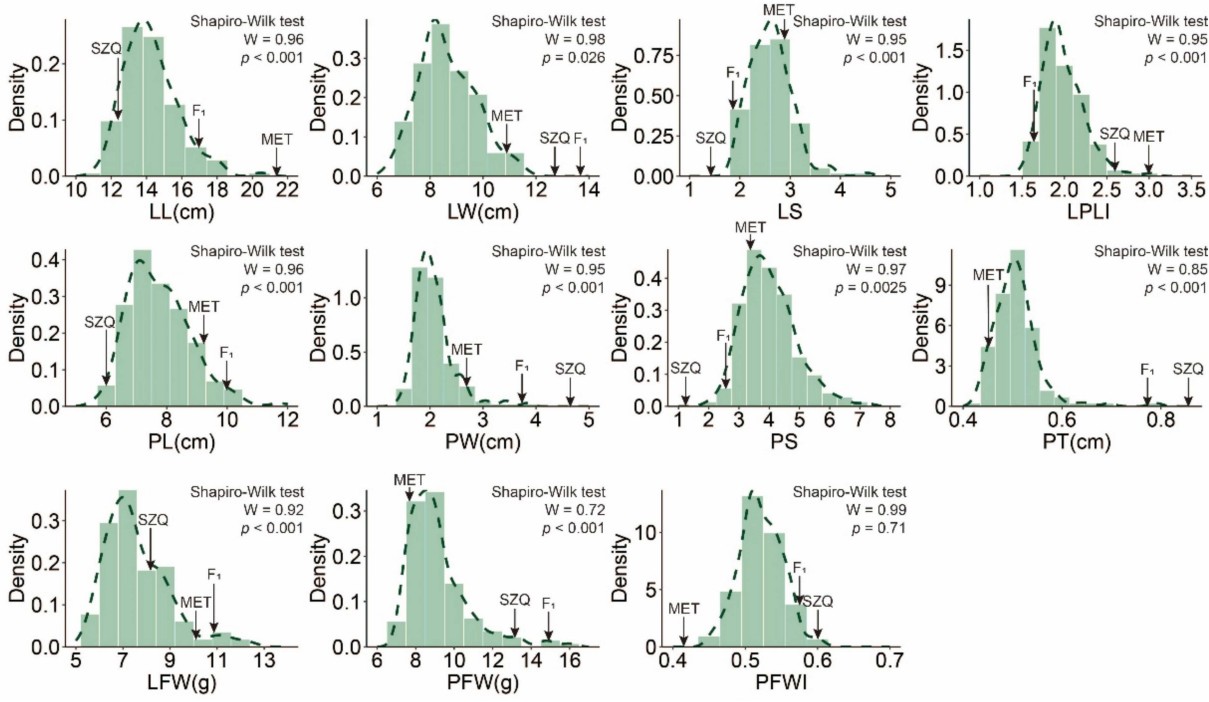

**Figure 3.** The distribution of 11 leaf-related traits in RIL populations. The results of the normality test (Shapiro–Wilk test) were displayed at the upper right corner of each histogram. Arrows indicated the 11 leaf-related traits of each parent and F1 hybrid. MET—parental lines 'Maertou'; SZQ—parental lines 'Suzhouqing'; $F_1$—the $F_1$ hybrid.

**Table 2.** Analysis of variance, variance component estimates, and heritability ($H^2$) for 11 leaf-related traits in RIL populations.

| Source of Variation | LL | LW | LS | LPLI | PL | PW | PS | PT | LFW | PFW | PFWI |
|---|---|---|---|---|---|---|---|---|---|---|---|
| Genotype (G) | 3.88 | 1.75 | 0.21 | 0.12 | 1.94 | 0.19 | 1.06 | 1.95 | 4.12 | 12.90 | 0.0019 |
| Location (L) | 1.71 | 0.83 | 0.00047 | 0.0022 | 0.70 | 0.06 | 0.00 | 0.70 | 9.08 | 20.50 | 0.00 |
| G × L | 4.28 | 1.59 | 0.042 | 0.17 | 2.73 | 0.083 | 0.56 | 2.72 | 6.22 | 45.40 | 0.0023 |
| Residual | 2.06 | 0.82 | 0.093 | 0.14 | 1.68 | 0.15 | 0.66 | 1.68 | 6.13 | 22.00 | 0.0030 |
| Heritability ($H^2$) | 0.43 | 0.46 | 0.56 | 0.28 | 0.33 | 0.41 | 0.46 | 0.33 | 0.25 | 0.19 | 0.25 |

Note: See Table 1 for trait abbreviations.

Some traits related to leaf shape containing LL, LW, PL, and PT exhibited significantly positive correlations with leaf weight traits LFW and PFW, while LS and PS showed significantly negative correlations with LFW; LS was positively correlated with PL and PS but negatively correlated with LPLI; PS was positively correlated with LS but negatively correlated with LW and LPLI. PT was positively correlated with LL and LW but negatively correlated with LS and PS. The highest correlation coefficient was observed between LFW and PFW (0.84). In RIL populations, leaves with elongated shapes often exhibit slender and lighter petioles, while oval leaves mostly had wide, thick, and heavy petioles, suggesting that the leaf shape and leaf weight may be controlled by the same genetic mechanisms.

However, there was no significant correlation between PW and other traits, suggesting the existence of a separate mechanism regulating PW (Figure 4).

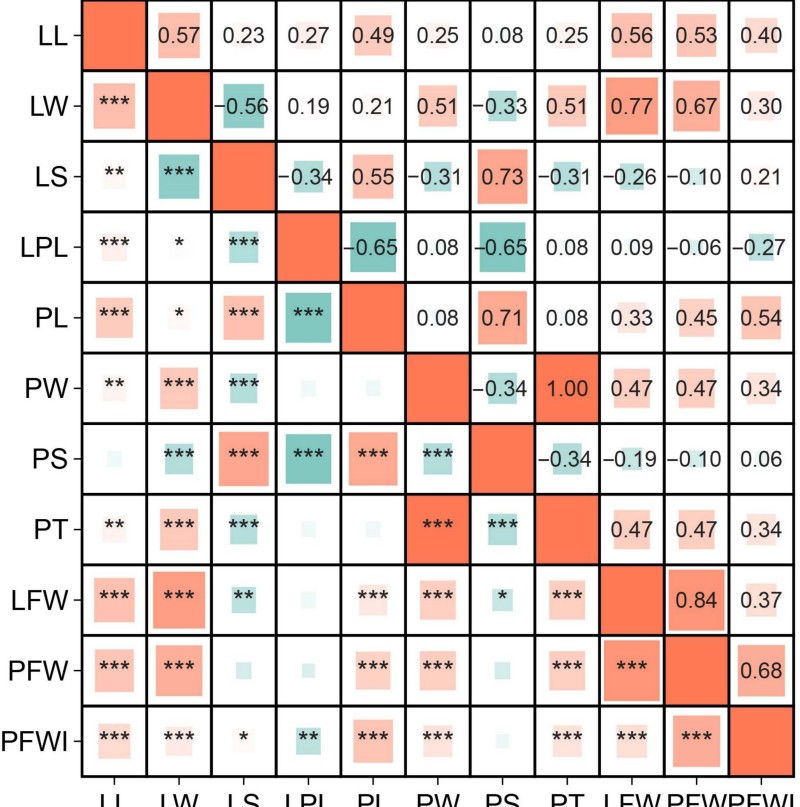

**Figure 4.** Correlation analysis of BLUPs for 11 leaf-related traits in the RIL population. (* $p \leq 0.05$, ** $p \leq 0.01$, *** $p \leq 0.001$).

### 3.2. Construction of Genetic Maps

The parental lines and the $F_{7:8}$ RIL population were re-sequenced and a total of 14.4 Gb and 331 Gb of data were obtained, respectively. For each line in the RIL population, we obtained 3.56 Gb of clean data, corresponding to an average depth of 9.44× (Table S2). A total of 7,365,439 SNP markers were selected and 5,325,599 SNP markers remained after quality control. A total of 845,023 SNP markers with the $aa \times bb$ pattern between parental lines were selected and 793,144 SNP markers with missing rates of less than 10% remained. The SNP markers with segregation distortion ($p < 0.0025$) and heterozygous rate >0.78% ($2^{-7}$) were filtered, and the remaining 1618 SNP candidates were adopted to estimate the linkage map (Table S3).

The genetic linkage map was constructed with 1618 SNPs. The total lengths of the genetic maps were 2294.1 cM (Tables 3, S3 and S4). The average number of markers per linkage group was 161.8 in the map, with average genetic distances per chromosome of 20.98 cM. These SNPs were evenly distributed across the 10 linkage groups.

**Table 3.** The summary of genetic linkage map.

| Linkage Group (Chr) | Chr Length | Detected SNP | Mapped SNP | Map Length (cM) | Average Spacing (cM) | Maximum Spacing (cM) |
|---|---|---|---|---|---|---|
| A01 | 38,126,442 | 722,047 | 170 | 292.4 | 1.70 | 22.7 |
| A02 | 33,817,099 | 753,934 | 164 | 227.2 | 1.40 | 10.6 |
| A03 | 41,214,576 | 779,968 | 103 | 283.7 | 2.80 | 34.6 |
| A04 | 23,891,077 | 530,781 | 51 | 130.0 | 2.60 | 17.4 |
| A05 | 42,748,514 | 794,914 | 108 | 227.9 | 2.10 | 20 |
| A06 | 43,549,656 | 897,224 | 485 | 349.7 | 0.70 | 7.4 |
| A07 | 29,241,068 | 616,408 | 60 | 62.1 | 1.10 | 14.5 |
| A08 | 23,546,603 | 559,964 | 144 | 203.7 | 1.40 | 15.7 |

**Table 3.** *Cont.*

| Linkage Group (Chr) | Chr Length | Detected SNP | Mapped SNP | Map Length (cM) | Average Spacing (cM) | Maximum Spacing (cM) |
|---|---|---|---|---|---|---|
| A09 | 64,520,918 | 1,276,206 | 238 | 338.5 | 1.40 | 50.5 |
| A10 | 23,121,789 | 433,993 | 95 | 178.9 | 1.90 | 16.4 |
| Average | | 736,543.9 | 161.8 | 229.4 | 1.40 | 21.0 |
| Summary | | 7,365,439 | 1618 | 2294.1 | 1.40 | 50.5 |

*3.3. QTL Mapping Analysis and Co-Localization of QTL*

A total of 27 significant QTLs were detected from 11 leaf-related traits using the CIM method (Tables 4 and S5 and Figure S2), 8 of which were detected jointly by the MQM method (Tables 4 and S6 and Figure S3).

**Table 4.** Quantitative trait loci (QTL) for leaf-related traits detected in RIL populations.

| Traits | Environment | QTL | Linkage Group | Peak Position (cM) | Genetic Interval (cM) | Physical Interval (bp) | LOD | Additive Effect | var% | Detected Method |
|---|---|---|---|---|---|---|---|---|---|---|
| LL | Baima | qLL.9.1 | A09 | 255 | 251.92–257.21 | 54,284,337–54,909,787 | 4.60 | 1.04 | 10.64% | CIM, MQM |
| | Baima | qLL.10.1 | A10 | 64 | 61.8–66.36 | 14,423,163–14,753,808 | 5.26 | 0.87 | 16.01% | CIM |
| | Huzhou | qLL.9.2 | A09 | 236 | 228.5–238.26 | 51,166,169–53,344,508 | 12.65 | 0.93 | 27.16% | CIM |
| | BLUP | qLL.9.2 | A09 | 234 | 228.5–238.26 | 51,166,169–53,344,508 | 9.26 | 0.51 | 21.03% | CIM |
| LW | Baima | qLW.9.1 | A09 | 259 | 258.2–258.895 | 54,979,140–55,034,347 | 258.55 | 0.61 | 8.86% | CIM |
| | Huzhou | qLW.1.1 | A01 | 26 | 19.48–35.20 | 989,993–1,872,968 | 4.87 | 0.58 | 9.80% | CIM |
| | Huzhou | qLW.6.1 | A06 | 306 | 302.56–308.15 | 39,464,218–40,133,203 | 5.86 | 0.59 | 11.59% | CIM, MQM |
| | Huzhou | qLW.9.2 | A09 | 204 | 202.4016–205.5288 | 47,132,468–47,510,459 | 10.15 | 0.68 | 21.16% | CIM, MQM |
| | BLUP | qLW.6.2 | A06 | 289 | 287.492–290.5821 | 37,994,746–37,994,793 | 4.91 | 0.28 | 11.21% | CIM, MQM |
| | BLUP | qLW.9.2 | A09 | 204 | 202.4016–205.5288 | 47,132,468–47,510,459 | 10.27 | 0.44 | 21.01% | CIM |
| LS | Baima | qLS.3.1 | A03 | 32 | 30.9917–32.67531 | 28,309,949–28,571,158 | 8.53 | −0.16 | 9.94% | CIM |
| | Baima | qLS.4.1 | A04 | 13 | 0–13.22851 | 4,135,569–4,221,964 | 6.75 | −0.20 | 14.55% | CIM |
| | Huzhou | qLS.4.1 | A04 | 16 | 10.62462–24.93579 | 4,676,839–13,960,200 | 7.94 | −0.22 | 16.86% | CIM |
| | BLUP | qLS.4.1 | A04 | 15 | 0–15.33323 | 4,052,077–4,221,964 | 6.88 | −0.18 | 19.82% | CIM |
| LPLI | Huzhou | qLPLI.6.1 | A06 | 114 | 111.6521–116.8251 | 9,114,145–10,250,895 | 6.62 | 0.19 | 16.28% | CIM, MQM |
| | BLUP | qLPLI.9.1 | A09 | 157 | 155.5247–158.1822 | 6,023,021–6,435,341 | 6.94 | −0.05 | 11.17% | CIM |
| PL | Huzhou | qPL.8.1 | A08 | 118 | 112.9256–120.7969 | 15,650,536–16,681,470 | 5.78 | −0.69 | 10.31% | CIM |
| | Huzhou | qPL.9.1 | A09 | 274 | 271.487–275.6874 | 55,973,892–56,514,679 | 10.0847876 | −2.14 | 22.56% | CIM, MQM |
| | BLUP | qPL.8.2 | A08 | 172 | 171.4044–172.7959 | 7,016,354–8,537,378 | 6.14 | −0.30 | 13.88% | CIM, MQM |
| | BLUP | qPL.9.2 | A09 | 245 | 243.6222–246.4983 | 55,241,258–55,583,460 | 10.63 | 0.46 | 20.28% | CIM |
| PW | Baima | qPW.1.1 | A01 | 237 | 234.4222–240.1689 | 33,706,868–34,001,075 | 8.88 | 0.15 | 16.35% | CIM |
| | Huzhou | qPW.1.2 | A01 | 150 | 140.0755–155.0015 | 28,431,036–30,881,834 | 8.49 | 0.16 | 20.26% | CIM |
| PS | Huzhou | qPS.1.1 | A01 | 227.5 | 224.3572–231.473 | 33,503,602–33,602,028 | 5.73 | −0.50 | 16.51% | CIM, MQM |
| | Huzhou | qPS.8.1 | A08 | 76.1 | 72.54486–79.48189 | 17,955,760–17,648,975 | 6.13 | −0.54 | 15.91% | CIM |
| | BLUP | qPS.1.1 | A01 | 228 | 224.3572–231.473 | 33,503,602–33,602,028 | 7.98 | −0.41 | 21.24% | CIM |
| | BLUP | qPS.8.2 | A08 | 126 | 120.7969–128.0647 | 14,932,889–16,681,470 | 7.08 | −0.33 | 13.44% | CIM |
| LFW | Baima | qLFW.9.1 | A09 | 259 | 258.1957–258.895 | 55,034,347–54,979,140 | 8.72 | 1.30 | 8.67% | CIM |
| | Huzhou | qLFW.9.2 | A09 | 74.9 | 71.86596–75.29802 | 1,527,172–1,736,911 | 9.69 | 0.44 | 24.59% | CIM |
| | BLUP | qLFW.9.1 | A09 | 259 | 71.86596–258.895 | 1,527,172–55,006,529 | 7.21 | 0.40 | 9.34% | CIM |
| PFW | Huzhou | qPFW.9.1 | A09 | 233 | 228.5003–234.9453 | 51,166,169–52,736,112 | 10.15 | 0.99 | 22.73% | CIM |
| | BLUP | qPFW.9.2 | A09 | 259 | 258.1957–258.895 | 54,979,140–55,034,347 | 20.55 | 0.53 | 7.52% | CIM |
| PFWI | Huzhou | qPFWI.9.1 | A09 | 277.49 | 272.9204–278.5876 | 56,320,818–56,815,511 | 8.28 | 0.03 | 17.27% | CIM, MQM |
| | Huzhou | qPFWI.10.1 | A10 | 5.38 | 0–15.32951 | 96,585–1,129,590 | 5.79 | 0.03 | 11.02% | CIM, MQM |
| | BLUP | qPFWI.9.1 | A09 | 280 | 275.6874–288.5661 | 56,514,702–57,563,461 | 6.78 | 0.01 | 16.51% | CIM, MQM |
| | BLUP | qPFWI.10.1 | A10 | 17 | 6.337199–18.792343 | 679,925–1,343,752 | 5.52 | 0.01 | 14.11% | CIM, MQM |

Note: See Table 1 for trait abbreviations.

For leaf size, 14 QTLs were co-located with LL, LW, PL, and PW, explaining 8.86–27.16% of the phenotypic variation. QTLs for LL, LW, and PW exhibited positive additive effects and QTLs for PL showed negative additive effects. The *qLW.6.1*, *qLW.9.2*, and *qPL.9.1* were jointly detected by CIM and MQM, explaining the highest phenotypic variation of LW and PL, respectively. For the shape of leaf and petiole, seven QTLs, accounting for 9.94–25.28% of the phenotypic variation, were detected for LS, LPLI, and PS, with negative additive effects except for *qLPLI.6.1*. The *qLPLI.6.1* and *qPS.8.1* were jointly detected by CIM and MQM, and *qLPLI.6.1* explained the highest phenotypic variation of LPLI. For leaf weight, six QTLs controlling LFW, PFW, and PFWI were detected, with positive additive effects and 7.52–24.59% of the phenotypic variation. The *qLFW9.1* and *qPFW9.2* overlapped at 54,979,140–55,034,347 bp on the A09 chromosome, in accordance with the significant positive correlation between LFW and PFW. The *qPFWI.9.1* and *qPFWI.10.1* were jointly detected by CIM and MQM and stably appeared in two environments; *qPFWI.9.1* was the highest phenotypic variation of PFWI.

Several QTLs for leaf size, leaf shape, and leaf weight were found co-localized with each other (Figure 5). The QTLs for PL was associated with PS, which co-related at 14,932,889–16,681,470 bp on the A08 chromosome, in accordance with the significant pos-

itive correlation between PL and PS. A total of 12 QTLs related to leaf shape and leaf weight were detected on chromosome A09. Among them, four QTLs associated with LL, LW, LFW, and PFW were detected and overlapped at 54,979,140–55,583,460 bp on the A09 chromosome, which was consistent with the significant correlation between these traits. Another pleiotropic QTL, co-related with LL and PFW, was located at 51,166,169–53,344,508 bp on the A09 chromosome, in accordance with the significant positive correlation between LL and PFW. The co-localization suggested the pleiotropy of the genomic region above affect multiple traits.

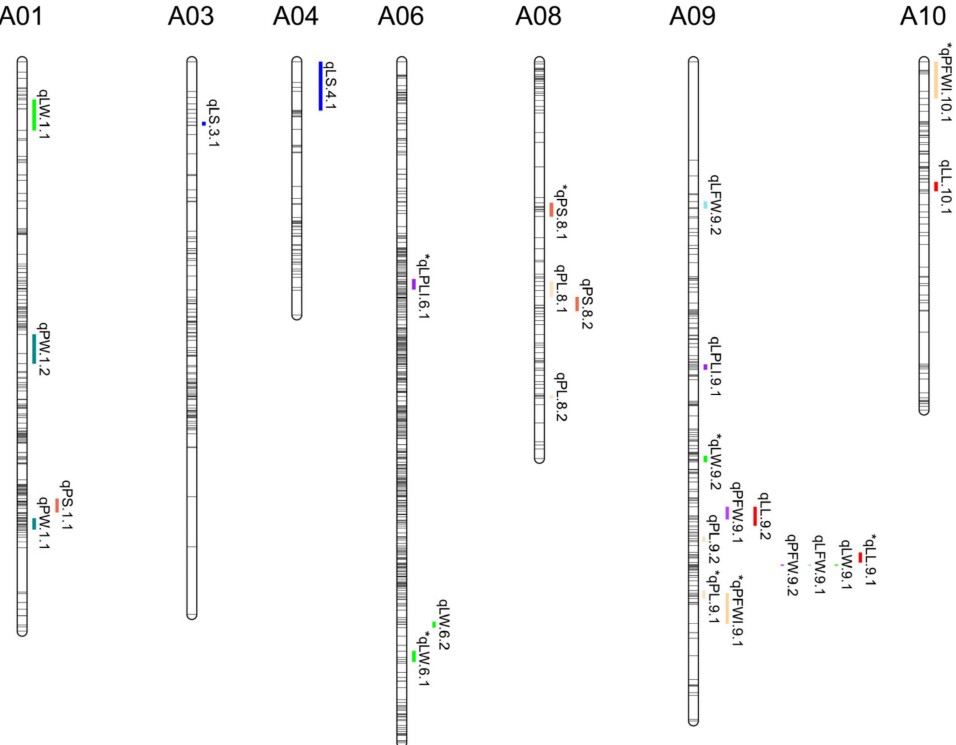

**Figure 5.** Distribution of the 11 leaf-related traits on the linkage map. The gray bars represent the 10 linkage groups on the linkage map and the colored bars represent QTL for the 11 leaf-related traits. Bar length indicates the 1.5-LOD supported interval of QTL. * represents the QTL detected by MQM and CIM method.

*3.4. Candidate Genes Analysis*

A total of 2737 genes were annotated in all significant QTL intervals. According to the gene description observed from TAIR (https://www.arabidopsis.org/, accessed on 20 March 2024), 30 genes involved in leaf development via cell proliferation and cell expansion, phytohormones, and other pathways were selected as putative genes (Table 5 and Table S7).

*BraC09g050650*, the homologue of *KRP2 (KIP-RELATED PROTEIN 2)*, encodes CDK (cyclin-dependent kinase) inhibitor (CKI), the negative regulator of cell division. *BraC09g051590*, the homologue of *GRF4 (GROWTH-REGULATING FACTOR 4)*, which is the growth-regulating factor that encodes the transcription activator. And these two genes were located on the common region of *qLL.9.2* and *qPFW.9.1*. *BraC09g057880*, as the homologue of *ARGOS* which is involved in lateral organ size controlling, was found at the common region of *qPL.9.1* and *qPFWI.9.1*. *BraC08g021310*, the homologue of *SAUR9* which encodes SAUR-like auxin-responsive protein, was found at the common region of *qPL.8.1* and *qPS8.2*. These genes located on the overlapped regions of co-localized QTLs were considered most likely the candidate genes. In addition, genes that were associated with leaf size and their development were therefore presumed to be putative candidate genes, including some genes affecting cell cycle machinery, such as *SMR6*, *APC2*, and *CCS52A1*; genes involved in

cell expansion, for example *APD6* and *HB12*; genes encoding members of SWI/SNF chromatin remodeling complex, including *SWI3C* and *SHH*; genes belonging to the DA1-EOD1 module including *DAR1*; genes involved in auxin and gibberellins pathways and genes regulating leaf development and size via other mechanisms identified in Arabidopsis, such as *RPS13A* [48].

**Table 5.** Candidate genes that may be associated with leaf-related traits.

| QTL | Gene ID in *Brassica campestris* | Homologue in Arabidopsis | Gene Name | References |
|---|---|---|---|---|
| *qLL.9.2* | *BraC09g051900* | *AT2G36985* | *ROT4* | Ikeuchi et al., 2011 [32] |
| *qLL.9.2* | *BraC09g051940* | *AT3G53250* | *SAUR57* | Spartz et al., 2012 [33]; Deng et al., 2019 [34] |
| *qPFW.9.1&qLL.9.2* | *BraC09g050650* | *AT3G50630* | *KRP2* | Cheng et al., 2013 [35] |
| *qPFW.9.1&qLL.9.2* | *BraC09g051590* | *AT3G52910* | *GRF4* | Lee et al., 2009 [36] |
| *qLW.1.1* | *BraC01g001950* | *AT4G36860* | *DAR1* | Peng et al., 2015 [37] |
| *qLW.1.1* | *BraC01g002470* | *AT4G36110* | *SAUR9* | Spartz et al., 2012 [33]; Deng et al., 2019 [34] |
| *qLW.1.1* | *BraC01g003510* | *AT4G34750* | *SAUR49* | Spartz et al., 2012 [33]; Deng et al., 2019 [34] |
| *qLS.4.1* | *BraC04g006480* | *AT4G22910* | *CCS52A1* | Baloban et al., 2013 [38] |
| *qLS.4.1* | *BraC04g007040* | *AT3G52910* | *GRF4* | Lee et al., 2009 [36] |
| *qLS.4.1* | *BraC04g007910* | *AT1G65800* | *RK2* | Sankaranarayanan et al., 2015 [39] |
| *qLS.4.1* | *BraC04g014700* | *AT2G04660* | *APC2* | Eloy et al., 2011 [40] |
| *qLS.4.1* | *BraC04g015220* | *AT5G40460* | *SMR6* | Michelle L et al., 2006 [41] |
| *qLPLI.6.1* | *BraC06g016490* | *AT1G21380* | *TOL3* | Barbara et al., 2013 [42] |
| *qLPLI.6.1* | *BraC06g016780* | *AT1G21700* | *SWI3C* | Vercruyssen et al., 2014 [15] |
| *qLPLI.6.1* | *BraC06g017160* | *AT1G21380* | *TOL3* | Barbara et al., 2013 [42] |
| *qLPLI.9.1* | *BraC09g010470* | *AT2G17800* | *ROP3* | Huang et al., 2014 [43] |
| *qLPLI.9.1* | *BraC09g010520* | *AT2G18010* | *SAUR10* | Spartz et al., 2012 [33]; Deng et al., 2019 [34] |
| *qPL.9.1&qPFWI.9.1* | *BraC09g057880* | *AT3G59900* | *ARGOS* | Wang et al., 2010 [44] |
| *qPL.9.2* | *BraC09g056020* | *AT3G57130* | *BOP1* | Hu et al., 2023 [45] |
| *qPL.8.1&qPS.8.2* | *BraC08g021310* | *AT4G36110* | *SAUR9* | Spartz et al., 2012 [33] |
| *qPW.1.2* | *BraC01g036690* | *AT3G20898* | *SMR13* | Michelle L et al., 2006 [41] |
| *qPS.8.1* | *BraC08g023640* | *AT4G38520* | *APD6* | Wong et al., 2019 [46] |
| *qPT.1.1* | *BraC01g041240* | *AT3G15540* | *IAA19* | Parameswari et al., 2016 [47] |
| *qLFW9.2* | *BraC09g002580* | *AT3G27630* | *SMR7* | Michelle L et al., 2006 [41] |
| *qPFWI.9.1* | *BraC09g058530* | *AT4G00100* | *RPS13A* | Ito et al., 2000 [48] |
| *qPFWI.9.1* | *BraC09g059400* | *AT3G61830* | *ARF18* | Zhang et al., 2021 [49] |
| *qPFWI.9.1* | *BraC09g059410* | *AT3G61840* | *ARF* | Zhang et al., 2021 [49] |
| *qPFWI.9.1* | *BraC09g059470* | *AT3G61890* | *HB12* | Hur et al., 2015 [50] |
| *qPFWI.9.1* | *BraC09g059720* | *AT3G62100* | *IAA30* | Parameswari et al., 2016 [47] |
| *qPFWI.10.1* | *BraC10g002600* | *AT1G04100* | *IAA10* | Parameswari et al., 2016 [47] |

## 4. Discussion

Leaves are the main photosynthetic and edible organ of non-heading Chinese cabbage. To explore the genetic and molecular basis controlling leaf size and leaf shape in *Brassica*, a genetic linkage map was constructed used a RIL population with 144 lines. QTL analysis revealed 27 significant QTLs on 7 chromosomes associated with 11 leaf-related traits.

Composite interval mapping (CIM) and multiple QTL mapping (MQM) are common methods for QTL mapping, fit for screening the additive effects. However, the QTL mapping power of MQM may be weaker than CIM in three QTL models [51,52]. In the present research, 27 and 8 significant QTLs were detected for leaf-related traits using the CIM and MQM methods, respectively, and the 8 QTLs detected by MQM were also detected by CIM. Apart from *qLL.9.1* and *qPS.8.1*, the six QTLs detected by MQM and CIM explained the highest phenotypic variation of the corresponding traits. In addition, we found that the results of the CIM were less stable than the results of the MQM. Some loci detected by CIM were only accidental and would disappear after repeated detections, and we so increased the number of repetitions, referring to the results of the MQM and the effects of peak markers (Figure S4) in RIL to select stable and reliable QTLs. To summarize, the CIM has more QTL mapping power but higher false positives, while the results of the MQM were more reliable but fewer QTL were detected. Therefore, it is helpful to use different methods to obtain the magnitude and reliable QTL for the target traits.

Previous studies illustrated multiple QTLs for leaf size and shape in *Brassica campestris*, some of which overlapped with QTLs in our research. The *qLS.4.1* overlapped with the QTL for TLL identified by Robert et al. using a RIL population [53]. The *qPW.1.2* co-located with the QTL for LL, LW, and TLL, detected by Xiao et al. using a DH population [6]. The overlapped region of *qLL.9.2* and *qPFW.9.1* contains the homologue of *KRP2*, which was

identified as a candidate gene for LW and LPLI [6]. These results suggested these QTLs are reliable and the genomic regions contain genes controlling leaf size.

The co-localization of QTLs and QTL clusters for different leaf traits has been reported in previous studies. Six QTLs for leaf area, petiole area, index of the leaf, LW, LFW, and PFW were co-located on chromosome 4 in *Brassica rapa* [8]. A total of 6 QTLs for LL, LW, and PL were mapped to a homologous region on chromosome 7 in *Brassica. oleracea* [21]. Several pleiotropic QTLs were identified, such as the *copQTL15* on A03 chromosome for leaf size (LW and LS) and leaf shape, and the *copQTL4* for LL, LW, and TLL, which co-located with the marker at 27.47 Mb on A01 chromosome in *Brassica rapa* [6]. The QTL for rosette leaf length and rosette leaf petiole length overlapped at the A05 chromosome in *Brassica rapa* [54]. In the present study, the following pleiotropic QTLs were found: four QTLs for leaf size and leaf weight were co-located on the A09 chromosome, two QTL for leaf size and leaf weight overlapped on the A09 chromosome, and two QTLs controlling petiole size and petiole shape overlapped on the A08 chromosome (Table 4), indicating the pleiotropy and close relation between leaf size, leaf shape, and leaf weight. The co-localized QTL regions could be a valuable resource for understanding the genetic basis of leaf size and leaf shape.

In previous studies, some QTLs were identified using RFLP, RAPD, and SSR markers with no specific physical location [21,55,56], so it was difficult to determine the relationship between these QTLs and the QTLs in this study. Compared with other loci with specific physical locations, the QTLs for LPLI, PS, and PFWI were first detected and co-located with corresponding traits. Furthermore, two pleiotropic regions on the A08 and A09 chromosomes, which contained multiple genes involved in auxin, cell proliferation, and expansion, were identified in this study, providing novel insights for gene cloning.

According to the gene annotation in 1.5-LOD support intervals, 30 genes involved in leaf size and development were identified as the possible candidate genes. It has been reported that these gene are involved in the regulation of leaf development in Arabidopsis. *BraC09g050650*, the homologue of *KRP2* in *Brassica rapa*, negatively regulates cell division [35]. *BraC09g051900*, the homologue of *ROT4*, acts as a regulator of leaf cell proliferation [32]. *BraC09g059470* and *BraC06g016780* are the homologues of *HB12* and *SWI3C*, respectively, which affect cell expansion [50] and cell number [15]. The homologue of *BraC09g058530* encodes a cytoplasmic ribosomal protein S13 (RPS13A) and is involved in early leaf development [48]. *BraC04g006480* is the homologue of *AtCCS52A* and the overexpression of *AtCCS52A* provided the change in leaf area [38]. The candidate genes will be further identified in the future and offer a genetic basis for regulating leaf-related traits in *B. campestris*.

## 5. Conclusions

In conclusion, a genetic linkage map was constructed based on the re-sequencing of the parental lines and RIL populations consisting of 144 lines. A total of 11 leaf-related traits were collected on two locations. A total of 27 significant QTLs on 7 chromosomes were associated with 11 leaf-related traits through the CIM and MQM methods and the candidate genes were annotated. The findings of this study will provide valuable information for further exploration of the genetic mechanisms of leaf shape, leaf size, and leaf development and lays a foundation for developing non-heading Chinese cabbage cultivars that are desirable to consumers.

**Supplementary Materials:** The following supporting information can be downloaded at: https://www.mdpi.com/article/10.3390/horticulturae10050529/s1, Figure S1: The distribution of 11 leaf-related traits in and RIL populations in Baima and Huzhou. Figure S2: Overview of the significant QTL for the 11 leaf-related traits identified in the RIL population using CIM method. Figure S3: Overview of the significant QTL for the 11 leaf-related traits identified in the RIL population using MQM method. Figure S4: Genotypic effects at peak marker locations of QTL for 11 leaf-related traits. Table S1: The descriptive statistics of five traits associated with leaf-related traits evaluated in RIL population. Table S2: Information of samples in RIL population and parent lines after re-sequenced. Table S3:

The information of genetic linkage map. Table S4: The summary of genetic linkage map. Table S5: Quantitative trait loci (QTL) for leaf-related traits detected in RIL population using CIM method. Table S6: Quantitative trait loci (QTL) for leaf-related traits detected in RIL population using MQM method. Table S7: Candidate genes that may be associated with leaf-related traits.

**Author Contributions:** Y.L. designed the study. T.Z., A.B. and X.W. conducted the experiments and analyzed the data. T.Z. wrote the manuscript. Y.L. and A.B. revised the manuscript. F.Z. and M.Y. helped collect the phenotypic data. Y.W., T.L., X.H. and Y.L. helped prepare the samples. All authors have read and agreed to the published version of the manuscript.

**Funding:** This study was supported by National Natural Science Foundation of China (32172565), Jiangsu Seed Industry Revitalization Project [JBGS (2021)015], and National Vegetable Industry Technology System (CARS-23-A-16).

**Data Availability Statement:** The raw sequence data generated during the current study are available in the publicly accessible National Center for Biotechnology Information (NCBI, https://www.ncbi.nlm.nih.gov/, accessed on 1 March 2024) database with accession number PRJNA1082622.

**Acknowledgments:** We thank anyone who helped us in this study.

**Conflicts of Interest:** The authors declare no conflicts of interest.

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
