# Peer review of "Genetic Mapping for Leaf Shape and Leaf Size in Non-Heading Chinese Cabbage by a RIL Population"

_horticulturae, doi:10.3390/horticulturae10050529_

Round 1
Reviewer 1 Report
Comments and Suggestions for Authors
1. The study is aimed to investigate the genetic basis of leaf shape and size in non-heading Chinese cabbage by constructing a genetic linkage map and by using a RIL population with 144 lines, which were developed by two varieties of this plant.
2. Few studies were carried out about this topic. Therefore, the aim of the work is well placed.
3. The contents of the manuscript are well described.
4. Figures and tables are clear and comprehensive.
However, I suggest some minor revisions:
· In the ‘Introduction’ section I suggest to add some general information at the beginning about Nonheading Chinese cabbage, such as its family its origin, grown cycle, and so on. Furthermore, here its scientific name (Brassica campestris L.) could be mentioned, instead it is cited for the first time in ‘Material and Methods’ section.
· Some previous studies about Brassica campestris L. including leaf investigation and its morphological traits were carried out for some time, such as of You et al., 2003, available at https://pubmed.ncbi.nlm.nih.gov/14986434/ . In the ‘Discussion section’, please underline better the new insights of the work respect to previous studies on the treated research topic.
· References could be improved and enriched with specific scientific works on the topic.
Comments on the Quality of English Language
Minor editing of English language required
Author Response
Response to Reviewer 1 Comments
|
||
1. Summary |
|
|
Thank you very much for taking the time to review this manuscript. Please find the detailed responses below and the corrections highlighted in the re-submitted files. |
||
2. Questions for General Evaluation |
Reviewer’s Evaluation |
Response and Revisions |
Does the introduction provide sufficient background and include all relevant references? |
Can be improved |
|
Are all the cited references relevant to the research? |
Can be improved |
|
Is the research design appropriate? |
Yes |
|
Are the methods adequately described? |
Yes |
|
Are the results clearly presented? |
Yes |
|
Are the conclusions supported by the results? |
Yes |
|
3. Point-by-point response to Comments and Suggestions for Authors |
||
Comments 1: In the ‘Introduction’ section I suggest to add some general information at the beginning about non-heading Chinese cabbage, such as its family its origin, grown cycle, and so on. Furthermore, here its scientific name (Brassica campestris L.) could be mentioned, instead it is cited for the first time in ‘Material and Methods’ section. |
||
Response 1: Thank you for pointing this out. We agree with this comment. Therefore, we have added general information about non-heading Chinese cabbage including its family, origin, grown cycle, and characters. This change can be found at Pg1, line #32-36. |
||
Comments 2: Some previous studies about Brassica campestris L. including leaf investigation and its morphological traits were carried out for some time, such as of You et al., 2003, available at https://pubmed.ncbi.nlm.nih.gov/14986434/. In the ‘Discussion section’, please underline better the new insights of the work respect to previous studies on the treated research topic. |
||
Response 2: Thank you for pointing this out. We agree with this comment. We have described the innovation point of this study. This change can be found at Pg15, line #333-339. |
||
Comments 3: References could be improved and enriched with specific scientific works on the topic. |
||
Response 3: Thank you for pointing this out. We have added the regulators controlling leaf size in ‘Introduction’ section and refined the literature of candidate genes for leaf shape and leaf size. This change can be found at Pg2, line #75-80 and Pg 13, line #276 (Table 5). |
||
4. Response to Comments on the Quality of English Language |
||
Point 1: Minor editing of English language required. |
||
Response 1: Thanks for your comments. We have carefully checked and corrected grammatical errors. This change can be found in manuscript. |
||
5. Additional clarifications |
||
|

Reviewer 2 Report
Comments and Suggestions for Authors
Manuscript ID: horticulturae-2966396
Type of manuscript: Article
Title: Genetic Mapping for Leaf Shape and Leaf Size in Non-heading Chinese Cabbage by a RIL Population
In this manuscript, authors developed a RIL population of Non-heading Chinese Cabbage, using two different genotypes, and they constructed a genetic linkage map and detected QTLs related to various leaf characteristics.
This work, in general, is very well conducted. But in my opinion, authors should attend to some details, so that it can be published:
1. Abstract. The scientific name of Non-heading Chinese Cabbage does not appear in the summary, in fact it is not mentioned until the materials and methods section. I consider it should be mentioned before, since this crop may have another common name in other countries.
2. Results: I consider that it would be desirable for the authors to include a graphic representation of the linkage map, in which all the QTL obtained were indicated (from the different environments and by the two methods used).
According to the above, I consider that the manuscript should be Accepted after minor revisions.
Author Response
Response to Reviewer 2 Comments
|
||
1. Summary |
|
|
Thank you very much for taking the time to review this manuscript. Please find the detailed responses below and the corrections highlighted in the re-submitted files. |
||
2. Questions for General Evaluation |
Reviewer’s Evaluation |
Response and Revisions |
Does the introduction provide sufficient background and include all relevant references? |
Can be improved |
|
Are all the cited references relevant to the research? |
Can be improved |
|
Is the research design appropriate? |
Yes |
|
Are the methods adequately described? |
Yes |
|
Are the results clearly presented? |
Yes |
|
Are the conclusions supported by the results? |
Yes |
|
3. Point-by-point response to Comments and Suggestions for Authors |
||
Comments 1: Abstract. The scientific name of Non-heading Chinese Cabbage does not appear in the summary, in fact it is not mentioned until the materials and methods section. I consider it should be mentioned before, since this crop may have another common name in other countries. |
||
Response 1: Thank you for pointing this out. We agree with this comment. Therefore, we have added scientific name of non-heading Chinese cabbage in abstract. This change can be found at Pg1, line #16. |
||
Comments 2: Results: I consider that it would be desirable for the authors to include a graphic representation of the linkage map, in which all the QTL obtained were indicated (from the different environments and by the two methods used). |
||
Response 2: Thank you for pointing this out. We agree with this comment. We have added a figure including linkage map and QTL. This change can be found at Pg11, line #252 (Figure 5). |
||
4. Response to Comments on the Quality of English Language |
||
Point 1: I am not qualified to assess the quality of English in this paper. |
||
Response 1: Thanks for your comments. We have carefully checked and corrected grammatical errors. This change can be found in manuscript. |
||
5. Additional clarifications |
||
|
